# Consistent Nest Site Selection by Turtles across Habitats with Varying Levels of Human Disturbance

**Molly Folkerts Caldwell [1],\*, Jorge E. López-Pérez [2], Daniel A. Warner [1] and Matthew E. Wolak [1]**

1    Department of Biological Sciences, Auburn University, Auburn, AL 36849, USA
2    Biology Department, Utah State University, Logan, UT 84322, USA
\*    Correspondence: mmf0009@auburn.edu

**Abstract:** Human disturbance impacts the breeding behavior of many species, and it is particularly important to understand how these human-caused changes affect vulnerable taxa, such as turtles. Habitat alteration can change the amount and quality of suitable nesting habitat, while human presence during nesting may influence nesting behavior. Consequently, both habitat alteration and human presence can influence the microhabitat that females choose for nesting. In the summer of 2019, we located emydid turtle nests in east-central Alabama, USA, in areas with varying levels of human disturbance (high, intermediate, low). We aimed to determine whether turtles selected nest sites based on a range of microhabitat variables comparing maternally selected natural nests to randomly chosen artificial nests. We also compared nest site choice across areas with different levels of human disturbance. Natural nests had less variance in canopy openness and average daily mean and minimum temperature than artificial nests, but microhabitat variables were similar across differing levels of disturbance. Additionally, we experimentally quantified nest predation across a natural to human-disturbed gradient. Nest predation rates were higher in areas with low and intermediate levels of disturbance than in areas with high human disturbance. Overall, these results show that turtles are not adjusting their choices of nest microhabitat when faced with anthropogenic change, suggesting that preserving certain natural microhabitat features will be critical for populations in human-disturbed areas.

**Keywords:** nesting behavior; nest predation; oviposition-site choice; *Trachemys scripta*; urbanization; yellow-bellied slider





## 1. Introduction

Urbanization dramatically changes the natural landscape and has numerous consequences on local climate, habitat structure, and biota. The damaging effects of urbanization on habitat quality and biodiversity are well documented [1–4] and can rapidly alter the behavior and survival of native species. A wide range of organismal responses to urbanization have been documented [5,6], including rapid acclimation, habituation, and even adaptation to human presence and human-modified environments [7]. For example, many mammal species have acclimated to urbanized habitats by shifting their behaviors, activity budgets and diet preferences [8]. Documenting these types of organismal responses to increasing urbanization is necessary to conserve wildlife populations and to determine ways to maintain key ecosystem functions [6].

Oviparous organisms typically rely on specific habitat characteristics for successful nesting. In most human-disturbed landscapes, however, habitat variables that are important for nest success (e.g., ground substrate, shade cover, temperature, predator densities) are heavily modified [9–11]. For example, urbanization can generate deviations from optimal thermal and hydric conditions in nests, leading to reduced offspring survival [12–15] and skewed population sex ratios for species with temperature-dependent sex determination (suggested by [16,17]). Urbanization can also alter nest-predator (e.g., racoons, dogs, and

cats) densities and activity patterns, leading to increased predation rates ([18–21], but see [22]). Conversely, predator activity or movement may decrease or remain concentrated in pockets of urbanized areas in ways that may reduce rates of nest predation [23–25].

The environmental effects described above imply that nest site choice is under selective pressure, because it links female behavioral traits to the survival, hence fitness, of her offspring [26,27]. Females typically select nest sites with abiotic conditions that enhance egg hatching success and positively affect offspring phenotypes [28,29]. Females also choose nest environments that minimize the risk of predation either to themselves or their offspring [30–34]. Accordingly, nest site choice encapsulates multiple issues in urban ecology because it (1) has important fitness consequences, (2) is influenced by available habitat and various biotic and abiotic factors, and (3) is altered in human-disturbed habitats [9,35–38]. Given the dramatic changes in habitat variables in areas with intense human activity, nesting females must either shift their choice of nest microhabitat or seek out pockets of preferred nest microhabitat across a human-disturbed landscape.

Turtles are a globally imperiled taxon [39] and are particularly susceptible to rapid environmental changes associated with urbanization due to a variety of factors associated with their life history (e.g., low survival during early life stages, delayed sexual maturity, environmental sex determination) [40]. Moreover, predation on turtle nests is common [41] and can potentially hinder recruitment into adult age classes [42]. Threats caused by urbanization also extend to later life stages, as adult turtles are highly vulnerable to predators [43] and other urban-associated mortality during nesting forays and other overland migrations (e.g., road mortality [44,45]). Thus, the impacts of environmental change due to urbanization on turtles has received considerable research attention [46–49]. However, relatively few studies of reptiles have examined how varying degrees of human activity and infrastructure affect nesting habitat and its consequences on nest site choice and nest predation rates [9,17,50].

In this study, we examined nest microhabitats chosen by female emydid turtles across areas with varying degrees of human disturbance and urbanization in east-central Alabama, USA, which is within a global biodiversity hotspot for turtles [51]. Emydid turtles are abundant across all our study areas, which vary in both the degree of human disturbance as well as in available nesting habitat. Given this observation, we hypothesized that females seek and use more specific nest microhabitats (e.g., temperature, shade cover) than what is available across the landscape. This aspect of our study was designed to determine whether turtles discriminate among abiotic factors when selecting a nest site. We also predicted that the level of human disturbance at our study areas would influence maternally selected microhabitat variables. Additionally, we quantified nest predation to provide insight into the role of this biotic factor in shaping nest success across natural to human-disturbed areas. Specifically, we performed an experiment with artificially constructed nests to determine if nest predation rates differ among our study areas with respect to the level of human disturbance. Consistent with our observations, we predicted that rates of nest predation would be highest in areas with less human disturbance.

## 2. Methods

### 2.1. Field Data Collection

We located emydid turtle nests surrounding 13 ponds in Lee and Macon counties, Alabama, from May to July 2019 (Table 1). Although our study areas contained nearby forested areas, we focused our surveys in open habitat at each area because numerous studies show that emydid turtles choose open habitat for nesting [29,52,53]. We visited all study areas regularly during the nesting season to maximize discovery of recent nests. We did not formally quantify effort by study area. However, we searched for nests on average six, thirteen, and three times per month for the high, intermediate, and low disturbance study areas, respectively. The difference in visitation rate to the different study area types is due to differences in the number and size of potential nesting areas: high disturbance study areas total approximately 20.3 ha across nine locations, intermediate areas total

approximately 24.6 ha across 16 locations, and low disturbance areas total approximately 1.3 ha across two locations. We found intact nests by observation of actively nesting females or by visual inspection of the ground for nests. Depredated nests, which were visually obvious as partially excavated cavities with eggshells scattered nearby, were also recorded during our survey. Depredated nests could have been from *Trachemys scripta*, *Pseudemys concinna*, *Chrysemys picta*, or *Terrapene carolina*. However, all nests for which the species could be confirmed were of *T. scripta*, the most common emydid turtle in our study areas. Here, we only report data from nests with obviously elliptical eggs characteristic of emydid turtles, as opposed to the spherical eggs of Chelydridae and Trionychidae.

**Table 1.** Study areas where turtle nesting was observed and where the nest predation experiments were performed. The coordinates for the pond on private property were omitted to maintain landowner privacy.

| Disturbance Level | Study Area | Pond Size (m$^2$) | Coordinates | Nesting Study vs. Nest Predation Experiment |
|---|---|---|---|---|
| High | Town Creek Park | 4561 | 32.582539, −85.476735 | Both |
| High | Kiesel Park | 742 | 32.587040, −85.542433 | Nesting study |
| High | Longleaf Villas | 3131 | 32.570633, −85.506619 | Nesting study |
| High | Agricultural Heritage Park | 8907 | 32.594622, −85.675574 | Predation experiment |
| Intermediate | Fisheries pond S10 | 11,558 | 32.669121, −85.508862 | Both |
| Intermediate | Fisheries pond S11 | 11,485 | 32.671127, −85.507211 | Both |
| Intermediate | Fisheries pond S2 | 7224 | 32.683346, −85.516154 | Nesting study |
| Intermediate | Fisheries pond S23 | 5600 | 32.678296, −85.517820 | Nesting study |
| Intermediate | Fisheries pond S24 | 7085 | 32.680441, −85.518099 | Nesting study |
| Intermediate | Fisheries pond S29 | 11,716 | 32.669498, −85.501004 | Nesting study |
| Intermediate | Fisheries pond S30 | 38,263 | 32.674933, −85.495792 | Nesting study |
| Intermediate | Fisheries pond S8 east | 5598 | 32.672734, −85.507651 | Nesting study |
| Intermediate | Fisheries pond S8 west | 37,512 | 32.672084, −85.509432 | Nesting study |
| Low | Tuskegee National Forest oxbow pond | 7342 | 32.439472, −85.635536 | Both |
| Low | Notasulga pond | 11,899 | - | Predation experiment |

Human disturbance and proximity to human infrastructure varied among our study areas, and as such, we ranked them as having high, intermediate, or low human disturbance (Table 1); we later confirmed these rankings with quantitative data on human census population size, amount of impervious surface, road density, and other variables (see details below). High disturbance areas were located at several city parks, as well as an apartment complex, in suburban areas of Auburn, Alabama. These areas were characterized as having infrastructure such as boardwalks and sidewalks adjacent to ponds and frequent human-related activity (e.g., pedestrians and pet dogs often swimming in ponds). Intermediate areas were located at Auburn University's EW Shell Fisheries Center in Auburn, Alabama, which has many ponds located in large grassy fields. These ponds experience periodic management such as grass mowing around the perimeters, but overall infrequent human visitation and little infrastructure. Low disturbance areas were a naturally formed oxbow pond located in Tuskegee National Forest, Macon county, Alabama and a private property pond located in Notasulga, Macon county, Alabama. These areas were far from human infrastructure and experienced very little, if any, human traffic. All study areas were in relatively close proximity to each other (within 30 km) and therefore eliminated the potential for confounding effects of geographic or climatic variation.

We measured several microhabitat characteristics at each intact and depredated nest. We measured the distance between each nest and the water using a measuring tape or wheel, in a straight line to the nearest shoreline of the closest body of water. We measured canopy openness by taking hemispherical photographs above the nest. We used a Nikon Coolpix L30 with a magnetic Zykkor fish eye 0.2X 180 degree lens. Before each photo, we oriented the camera facing north and then set the camera facing lens up directly on top of

the nest. We analyzed the photographs with GapLight Analyzer software to obtain percent canopy openness values [54]. We measured the slope of the ground surrounding the nest using the Apple Measure application.

To compare nest microhabitat to that available across the general landscape, we measured the same microhabitat variables described above at randomly selected artificial nest sites around each natural nest. We identified artificial nest site locations using a random number generator to obtain values between 1 and 10, indicating distance in meters from the natural nest, and values between 0 and 360, indicating the cardinal direction from the natural nest. Three artificial nests were created per natural nest. Henceforth, we refer to maternally selected nests as "natural nests" and randomly selected sites as "artificial nests".

We placed an iButton temperature logger inside nest cavities (for natural nests and a subset of artificial nests; $n = 47$ total) for the duration of the nesting season. Early in the season, we placed an iButton in every artificial nest. Later in the season, we placed an iButton in one of every three artificial nests, due to time limitations. iButtons were buried at the approximate depth of the cavity of the natural nest and at the same depth for the associated artificial nests. Temperature was recorded hourly to the nearest 0.5 °C. We trimmed iButton data to a total of 67 days (the average incubation length for *T. scripta* eggs at 28 °C; unpublished data), starting on the day after the iButton was deployed. Average daily maximum, mean, and minimum temperatures were obtained from all iButtons. We then averaged these values across all 67 days of data to obtain a single average daily maximum, mean, and minimum temperature value for each natural and artificial nest. Additionally, the average daily temperature range value for each iButton (maximum–minimum) was calculated. Each of these temperature variables was then used as a microhabitat variable in our analyses.

### 2.2. Nest Predation Experiment

We conducted an experiment to assess variation in nest predation across different levels of human disturbance at our study areas. Our experiment was performed at ponds of similar size at two areas with high human disturbance, two with intermediate disturbance, and two with low disturbance (Table 1).

We measured the perimeter of each pond that contained typical nesting habitat (open, grassy), placed a flag at each 10 m section, and assigned each section a number. Five sections (marked with flags) were selected using a random number generator; an artificial nest was constructed at each section at 3 m from the edge of the pond. Artificial nests had a cavity similar to that of a natural nest and consisted of a chicken egg buried in the soil at 10 cm depth (comparable to the depth of an emydid turtle nest). Past studies have successfully used bird eggs to quantify rates of predation on turtle nests [21,22,55–58]. We moistened the disturbed soil with pond water to simulate when female turtles release water from their bladder before nesting. Because nest predators rely on olfactory and visual cues to locate turtle nests [59], we reasoned that disturbed soil and the pond water provided realistic cues that simulated those of natural nests [60]. To discreetly mark the location of each artificial nest, we removed the flags and placed two wooden dowels one meter away from each nest. After burying eggs, we checked each artificial nest once every 24–48 h for a total of 72 h (as most predation occurs within this timeframe [41,61,62]; but see [63]). On each visit, evidence of predation, such as an excavated cavity or the presence of eggshells, was recorded and remaining eggs were removed after 72 h. This process was performed 3 times at each location, totaling 90 eggs buried across all six study areas. We randomly selected different sections of the pond for each repetition and waited at least 48 h before burying a new set of eggs at each section.

### 2.3. Quantifying Urbanization

To quantify the level of human disturbance at each study area, we downloaded data from the United States Environmental Protection Agency StreamCat Dataset that contains standardized measures of natural and anthropogenic features of streams, their catchments, and watersheds [64]. For each pond where we measured nest microhab-

itat or conducted the predation experiment, we used the WATERS GeoViewer (https://www.epa.gov/waterdata/waters-geoviewer (accessed on 20 September 2022)) to select the closest stream segment to the ponds at our study areas and downloaded the StreamCat data associated with the catchment area draining into that segment. Five variables were extracted from the watershed dataset: mean imperviousness of anthropogenic surfaces within catchment, percentage of local catchment area classified as developed, average density of roads per square kilometer, mean of all housing units per square kilometer, and mean of 2010 census population per square kilometer. For ponds that were equidistant to two segments, we took the average of the two segments for each variable. We performed a principal components analysis (PCA) on these five variables across each of our study areas. The first principal component (PC1) explained 88.62% of the variation in the data and was used in our subsequent analyses as a continuous variable associated with the level of human disturbance. We multiplied PC1 scores by negative one to facilitate ease of interpretation (Table 2) so that positive PC1 scores indicated high human disturbance, while negative scores were indicative of more natural areas with low human disturbance. Importantly, because our original classification of study areas as "high disturbance" vs. "intermediate" vs. "low disturbance" were reflected in the results of the PCA (Figure 1), some of our analyses also used the study areas as their original designations (rather than use PC scores). Although our sample of nests was low in the high disturbance study area (see Results), we still distinguished this study area type from the others due to dramatically different surroundings, infrastructure, and levels of human activity. We will refer to the original designations as "study area type" and to PC1 as "urbanization level".

**Table 2.** Loadings and proportion of variance explained on different principal component (PC) axes from a principal components analysis.

|  | PC1 | PC2 | PC3 | PC4 | PC5 |
|---|---|---|---|---|---|
| Impervious surface | 0.4624 | 0.0363 | −0.6312 | −0.6137 | −0.0988 |
| Developed area within catchment | 0.4732 | 0.0675 | −0.0300 | −0.2563 | 0.8396 |
| Density of roads | 0.4045 | 0.7692 | 0.4165 | 0.0355 | −0.2641 |
| Housing units | 0.4270 | −0.5903 | 0.6012 | 0.3196 | −0.0742 |
| Human population | 0.4649 | −0.2319 | −0.2562 | −0.6740 | −0.4583 |
| Proportion of variance explained | 0.8862 | 0.08643 | 0.0215 | 0.0046 | 0.0011 |

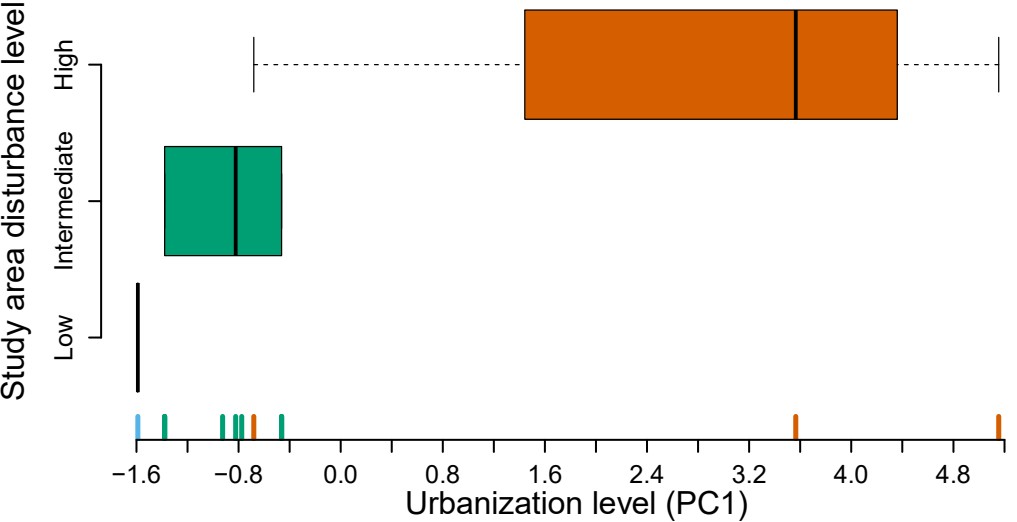

**Figure 1.** Level of human disturbance across study areas. Initial designation of human disturbance included three categories based on our observations (high, intermediate, low levels of disturbance). Principal component axis 1 (PC1) combines different metrics of human disturbance across study areas (see Table 2). Colored ticks along the *x*-axis indicate actual PC1 scores of each study area (some study areas within a human disturbance category have the same PC1 score).

### 2.4. Statistical Analysis

To broadly characterize high, intermediate, and low human disturbance study area microhabitats without respect to turtle nest site choice, we built linear models with a dataset that included artificial nests only, as these randomly located sites should provide a general description of the habitat at each study area. We used six models, each with a microhabitat variable (listed in Table 3) as the dependent variable, and study area type as a categorical fixed effect. We also included iButton depth as a continuous covariate in models that included a temperature-dependent variable. Distance from water was not included in this analysis since this variable is only meaningful in comparison with natural nests.

**Table 3.** Test statistics for comparisons of randomly selected locations (i.e., artificial nests) across three different types of study areas (high, intermediate, and low levels of human disturbance). Comparisons of temperature variables included iButton depth as an additional independent variable. For comparisons among study area types, effect sizes ($\beta$) were calculated using the low human disturbance area as the reference. Standard errors are indicated in parentheses, and statistically significant *p* values are in bold font.

| Dependent Variables | Study Area Type | iButton Depth |
|---|---|---|
| Slope<br>Intermediate disturbance<br>High disturbance | $F_{2,124}$ = 5.792, ***p* = 0.004**<br>$\beta$ = 5.106 (1.602), ***p* = 0.002**<br>$\beta$ = 6.233 (2.707), ***p* = 0.023** | - |
| Canopy openness (%)<br>Intermediate disturbance<br>High disturbance | $F_{2,126}$ = 4.573, ***p* = 0.012**<br>$\beta$ = 7.852 (4.238), *p* = 0.066<br>$\beta$ = −10.898 (7.197), *p* = 0.133 | - |
| Average daily mean temperature<br>Intermediate disturbance<br>High disturbance | $F_{2,19}$ = 1.946, *p* = 0.170<br>$\beta$ = −1.005 (1.168), *p* = 0.400<br>$\beta$ = 1.179 (2.272), *p* = 0.610 | $\beta$ = −1.778 (1.246)<br>*p* = 0.170 |
| Average daily maximum temperature<br>Intermediate disturbance<br>High disturbance | $F_{2,19}$ = 3.565, ***p* = 0.048**<br>$\beta$ = −2.627 (2.076), *p* = 0.221<br>$\beta$ = 2.263 (4.039), *p* = 0.582 | $\beta$ = −3.877 (2.214)<br>*p* = 0.096 |
| Average daily minimum temperature<br>Intermediate disturbance<br>High disturbance | $F_{2,19}$ = 0.131, *p* = 0.878<br>$\beta$ = −0.036 (0.645), *p* = 0.956<br>$\beta$ = 0.588 (1.256), *p* = 0.645 | $\beta$ = −0.417 (0.688)<br>*p* = 0.552 |
| Average daily temperature range<br>Intermediate disturbance<br>High disturbance | $F_{2,19}$ = 5.587, ***p* = 0.012**<br>$\beta$ = −2.591 (1.573), *p* = 0.116<br>$\beta$ = 1.675 (3.061), *p* = 0.590 | $\beta$ = −3.460 (1.678)<br>*p* = 0.053 |

To determine if females discriminate among abiotic factors when selecting nest sites, and whether microhabitat measures varied with level of disturbance, we included a set of seven linear mixed-effects models, each with a microhabitat variable as the dependent variable. We included nest type (natural vs. artificial nest) as a categorical fixed effect, urbanization level (PC1 described above) as a continuous covariate, and the interaction between those two variables. For models of temperature-dependent variables, a continuous covariate of iButton depth (mean centered and standard deviation scaled) was also included. Statistical significance of individual fixed effect terms was evaluated with an analysis of variance implementing incremental sums of squares. Nest cluster (a natural nest and its associated artificial nests) was assigned as a random effect in each model to account for the non-independence of nests within clusters. To determine whether female turtles selected nest sites with more or less variance in microhabitat measures, we estimated separate residual variances for natural and artificial nests. We used the asreml package [65] in R for each model and obtained 95% confidence intervals for the residual variances using profile likelihoods [66] implemented in the nadiv package [67]. A likelihood ratio test was used to evaluate the statistical null hypothesis that there is no difference in residual variance between natural and artificial nests.

To determine if nest predation rate was associated with study area type, we performed a Pearson's Chi-square test for independence, using the numbers of artificial nests that were depredated and the number that survived at each of the three study area types (high, intermediate, low disturbance). All statistical analyses were performed in R, version 4.2.1 [68].

## 3. Results

Slope of the ground and canopy openness measured at artificial nests (i.e., random locations representative of the landscape at each study area) varied with the level of human disturbance (Table 3). Ponds in more human-disturbed areas had steeper terrain and less canopy openness than those in natural areas. Mean and minimum ground temperatures did not change substantially with the level of human disturbance, but the daily maximum and daily range of ground temperature decreased with increasing human disturbance (Table 3).

### 3.1. Nest Site Choice

A total of 88 nests (11 intact, 77 depredated) were located during the study. Most nests ($n$ = 84) were at intermediate or low disturbance areas, and only four were found in high disturbance areas. Due to logistical limitations, microhabitat data were collected on a subset of natural nests and their associated artificial nests ($n$ = 43; 4 high, 25 intermediate, 14 low), and all results reported are from this set of nests. In addition, due to failure or loss of some iButtons, temperature data were available for 20 natural nests and 27 artificial nests.

Microhabitat variables measured at natural nests generally followed the same trends as those in artificial nests (Figure 2). Slope, daily mean, and daily maximum temperature were marginally significantly related to the level of urbanization (Table 4), and these relationships did not differ between natural and artificial nests. Nests in more urban areas were on steeper slopes than in natural areas (Figure 2B). Average nest temperature at natural areas (29.9 °C, $n$ = 11) was about 2 °C warmer than that at intermediate areas (28.0 °C, $n$ = 8), and daily maximum temperatures followed a similar trend; higher maxima in natural areas as compared to intermediate and high human disturbance locations (Figure 2). Although temperature data from only one natural nest iButton were available at a high human disturbance area, this trend was still observed when we compared just the intermediate to natural areas. The distance of natural nests to the nearest pond ranged from 2 to 241 m, with 88% of nests being within 50 m of a pond; this pattern did not vary with the level of human disturbance.

Although nest site microhabitat characteristics did not differ on average between natural and artificial nests, natural nest sites generally exhibited less variance compared to artificial nests. For instance, there was significantly ($p < 0.05$) less among-nest variance of natural nests in their distance to water, canopy openness, and average daily mean and minimum temperatures. However, the variance of daily temperature range of natural nests was greater than that of artificial nests. For other microhabitat measures, the best fit model indicated that residual variance was equal between natural and artificial nests.

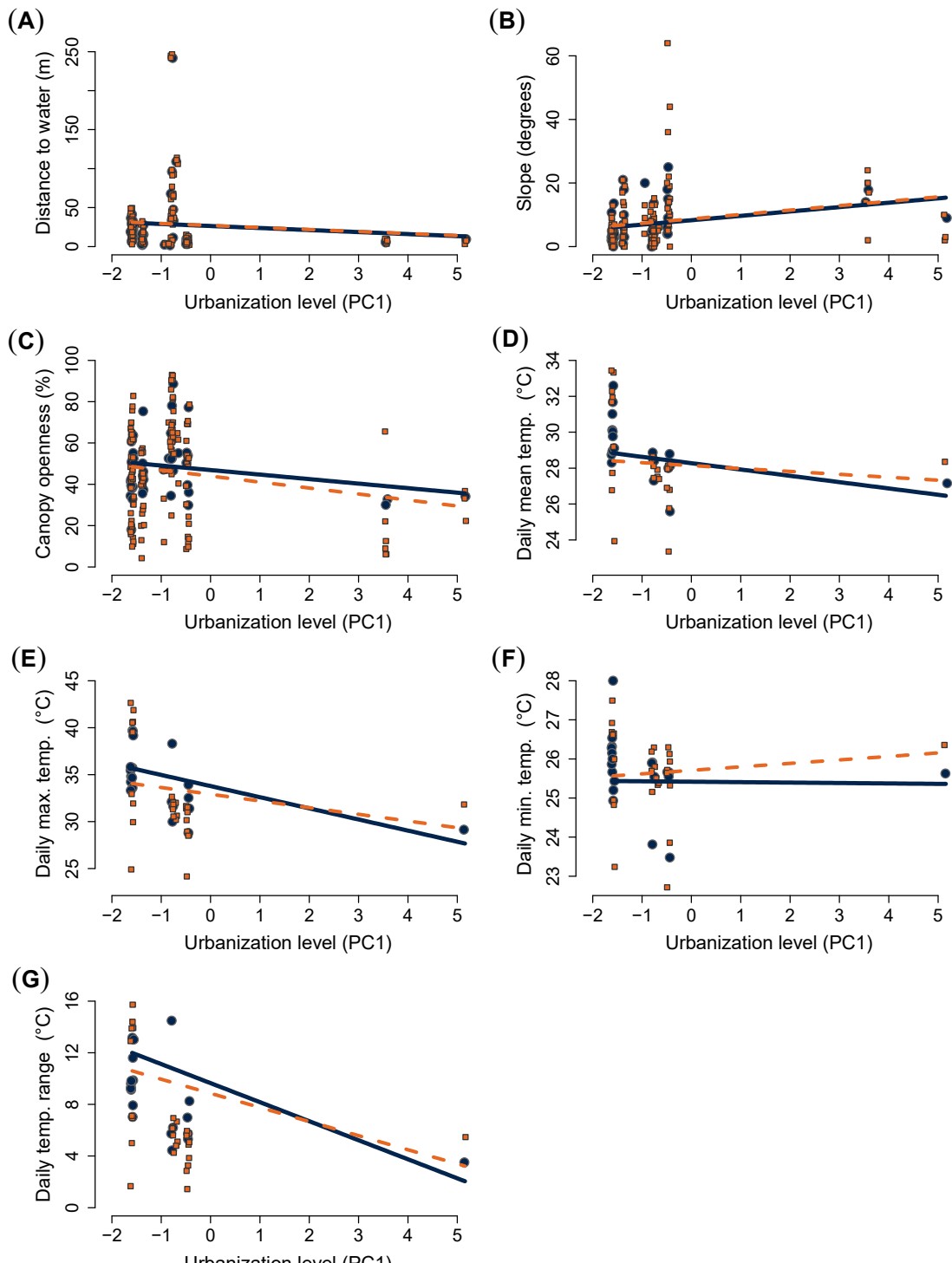

**Figure 2.** Effects of urbanization level for natural (blue circles, solid lines) and artificial (orange squares, dashed lines) nests on microhabitat variables. (**A**) Distance of nests from the edge of water of the nearest pond. (**B**) Ground slope around nest sites. (**C**) Canopy openness over nests. (**D**) Daily mean nest temperature. (**E**) Daily maximum nest temperature. (**F**) Daily minimum nest temperature. (**G**) Daily temperature range.

**Table 4.** Effects of urbanization, nest type (natural vs. artificial), and their interaction on microhabitat variables. iButton depth was only included in the analyses of temperature data. The effect size for nest type was calculated with the natural nest as the intercept/reference. The rightmost two columns indicate whether residual variance differed between natural (N) and artificial (A) nests, and if so, what the two separate variances are (natural nests listed first, artificial nests second). Estimates and either standard errors or 95% confidence intervals are indicated in parentheses. For residual variances, we report lower confidence interval limits of 0 when the CIs cannot exclude zero. Statistically significant *p* values are in bold font.

| | Urbanization Level (PC1) | Nest Type | Urbanization Level x Nest Type Interaction | iButton Depth | Residual Variances | Equal Variance Likelihood Ratio Test |
|---|---|---|---|---|---|---|
| Distance from water | $\beta$ = −2.589 (4.504) <br> *p* = 0.567 | $\beta$ = 0.625 (0.523) <br> *p* = 0.186 | $\beta$ = 0.020 (0.328) <br> *p* = 0.951 | - | N = 4.875 (2.331 to 8.453) <br> A = 12.889 (10.507 to 16.048) | $\lambda_1$ = 5.025 <br> ***p* = 0.012** |
| Slope | $\beta$ = 1.377 (0.831) <br> ***p* = 0.058** | $\beta$ = 0.428 (0.898) <br> *p* = 0.594 | $\beta$ = 0.006 (0.562) <br> *p* = 0.991 | - | N = 19.166 (12.147 to 29.477) <br> A = 23.457 (18.740 to 29.871) | $\lambda_1$ = 0.263 <br> *p* = 0.304 |
| % canopy openness | $\beta$ = −2.189 (1.826) <br> *p* = 0.124 | $\beta$ = −2.931 (2.259) <br> *p* = 0.228 | $\beta$ = −0.704 (1.415) <br> *p* = 0.619 | - | N = 90.671 (49.070 to 153.127) <br> A = 241.750 (191.827 to 306.462) | $\lambda_1$ = 4.712 <br> ***p* = 0.015** |
| Average daily mean temp. | $\beta$ = −0.339 (0.372) <br> *p* = 0.095 | $\beta$ = −0.134 (0.409) <br> *p* = 0.522 | $\beta$ = 0.189 (0.262) <br> *p* = 0.470 | $\beta$ = −0.021 (0.707) <br> *p* = 0.973 | N = 0.029 (0 to 1.228) <br> A = 2.659 (1.530 to 4.302) | $\lambda_1$ = 2.781 <br> ***p* = 0.048** |
| Average daily maximum temp. | $\beta$ = −0.333 (0.819) <br> *p* = 0.077 | $\beta$ = −0.871 (0.962) <br> *p* = 0.150 | $\beta$ = 0.472 (0.571) <br> *p* = 0.408 | $\beta$ = −1.435 (1.1494) <br> *p* = 0.335 | N = 3.777 (0 to 20.031) <br> A = 8.985 (1.899 to 17.140) | $\lambda_1$ = 0.321 <br> *p* = 0.286 |
| Average daily minimum temp. | $\beta$ = 0.032 (0.232) <br> *p* = 0.994 | $\beta$ = 0.291 (0.183) <br> *p* = 0.174 | $\beta$ = 0.100 (0.115) <br> *p* = 0.385 | $\beta$ = −0.072 (0.437) <br> *p* = 0.865 | N = 0.008 (0 to 0.279) <br> A = 0.512 (0.317 to 0.835) | $\lambda_1$ = 3.064 <br> ***p* = 0.040** |
| Average daily temp. range | $\beta$ = 0.482 (1.002) <br> *p* = 0.222 | $\beta$ = −0.793 (1.051) <br> *p* = 0.241 | $\beta$ = 0.382 (0.578) <br> *p* = 0.508 | $\beta$ = −3.296 (1.605) <br> ***p* = 0.040** | N = 11.348 (3.224 to 23.909) <br> A = 1.751 (0.828 to 7.223) | $\lambda_1$ = 1.205 <br> *p* = 0.136 |

### *3.2. Nest Predation Experiment*

Of the 90 artificial nests, we documented six instances of nest predation, and nest predation rate was significantly greater in natural areas than the intermediate and high disturbance areas (chi-square = 7.5, *p* = 0.045). Five of the six nests depredated were at natural areas (17% predation rate), and the remaining was at an intermediate disturbance area; no predation occurred in high disturbance areas. Out of the five nests depredated in natural areas, four occurred at the oxbow pond in Tuskegee National Forest, while one occurred at the private property pond in Notasulga, AL. Similarly, during our nest surveys, more depredated nests were observed in low (*n* = 27) and intermediate (*n* = 49) human disturbance study areas than at high disturbance areas (*n* = 1).

### 4. Discussion

Human activity has altered natural landscapes in ways that have dramatic effects on wildlife populations. These effects of habitat alteration may be particularly pronounced when they directly impact habitat or other environmental factors that are important for reproduction [69,70], such as nesting areas of oviparous species. In this study, we quantified variation in nesting habitat for turtles, maternal nesting behaviors, and nest predation rates across a range of areas that vary in the level of human disturbance. We found that nesting landscapes in areas with greater human disturbance had steeper slopes and reduced canopy openness than in areas with less human disturbance. The reduced variance in some microhabitat variables for natural nests suggests that female turtles choose microhabitats with specific abiotic conditions, and these patterns remained consistent across the study. Nest predation also varied among study locations and was substantially lower in areas with high levels of human disturbance. These patterns illustrate that while human activities alter natural habitats, female turtles are still capable of finding nest sites with similar microhabitats across different levels of disturbance.

While the distance of nests to water and average canopy over nests tended to reflect what was available across study areas, female turtles nested in a narrower range of these variables than what was measured at artificial nest sites, suggesting that females are selective for distance from the water's edge and canopy when choosing a nest site. However, we detected minimal differences for maternally selected nest microhabitat characteristics among different study area types on an urban gradient, suggesting that females discriminate among abiotic factors when selecting nest sites but generally select sites with similar characteristics at each level of human disturbance. Nesting turtles may not adjust their nest site choice when faced with anthropogenic change, highlighting the need for protecting areas with suitable habitat for nesting and egg incubation, particularly areas that are altered by humans. However, more investigation is needed to determine whether turtles are capable of adjusting their nesting strategy in these areas or whether they were able to find appropriate nest sites without substantial changes to their nesting behaviors.

Nest temperatures were relatively high in natural areas compared to high and intermediate disturbance areas. This result contrasts with those of studies on the urban heat island effect whereby urbanized areas are considerably warmer than surrounding natural areas due to a variety of factors (e.g., increased heat absorbing surface, decreased tree cover; [71]). Notably, our study areas with the highest levels of human disturbance were more suburban than urban and still contained substantial amounts of greenspace (i.e., city parks) that would reduce the likelihood of elevated temperatures comparable to those in large cities. The range of human disturbance in our study does not extend to the extremes seen in large cities [72]. Consequently, variation in temperature across our study areas is more subtle, especially with the limited temperature data collected for nests in our most human-disturbed areas. Nevertheless, we detected increased nest temperatures in natural areas, which could be driven by substrate differences; all nests at the natural location had pebbly substrate that may absorb more heat than the grassy/soil substrate at our more urban study areas. These results highlight the importance of considering local microhabitat characteristics in driving thermal patterns across urban to natural gradients, as major assumptions in urban ecology (e.g., urban areas are always relatively warm; [71]) may not always be met. Relying solely on these broad assumptions, especially when the gradient is relatively shallow, could impact the success of management efforts for wildlife species.

The artificial ponds at our high and intermediate human disturbance areas were surrounded by steeper terrain than ponds in natural areas, which is a common characteristic of human-made wetlands that were carved out by construction equipment. Consequently, turtles nested on relatively sloped ground at these areas, which reflects this feature of human-made ponds, rather than being indicative of turtle nest site choice. Nevertheless, a high occurrence of steep slopes in nesting areas could affect ambient conditions of the nest that have important consequences on embryo development. For example, sloped banks around the pond will affect radiant heating from the sun (especially on south-facing slopes in the northern hemisphere) and have impacts on nest temperature in ways that influence critical aspects of development [73,74]. Sloped banks could also influence water run-off and decrease the moisture absorbed in the soil at a nest site; these impacts on nest moisture could also influence development of turtle embryos [75,76]. Thus, while properly managed artificial wetlands can support healthy wildlife populations [77], the surrounding features of artificial ponds are often very different from that of natural ponds and may influence nesting behavior and embryo ecology of turtles in unique ways.

The abundance of nests was relatively low in areas with high human disturbance, despite a high abundance of turtles observed in ponds at these study areas. Over 70 adult turtles can readily be counted (within 1–2 min) in the ponds at city parks and apartment complexes (pers. obs.), whereas few, if any, turtles are observed within this short timeframe at our study areas with low human disturbance; these observations are mostly driven by turtle habituation to humans, as they are often fed by visitors at city parks (as seen in other wildlife; [78]) and not at the undisturbed areas. Given the high apparent densities of adult turtles in city parks, the lack of nest sites was unexpected, which may be due to

several factors. First, females may choose nest sites far outside of our survey areas. While it is possible that females may travel farther distances across land in human-disturbed areas, we argue that this is unlikely because of barriers around the parks (roads, residential areas). Alternatively, females may nest in heavily shaded forest patches surrounding the city parks where we did not search, but a large amount of literature consistently shows that emydid turtles select open habitat for nesting [9,29,79], which was abundant at our human-disturbed study areas. Second, human disturbance may have caused diel shifts toward nocturnal nesting activity outside of our survey hours. Adaptive shifts toward nocturnal nesting activity at human-disturbed areas is unlikely given the long lifespans of turtles and the relatively short time since establishment of our city park study areas. In addition, other studies provide no evidence that emydid turtles shift the diel timing of nesting activity, as they readily nest during the day in areas with high human activity [80]. The lack of depredated nests at city parks also implies little nesting activity outside the times of our surveys. A third explanation involves potential physiological effects of human disturbance that may inhibit reproduction. Trapping efforts (in 2021 and 2022) show that almost no females in city park ponds are gravid during the reproductive season and smaller, juvenile turtles are rarely observed (unpubl. data), indicating low recruitment. Although urbanization has not previously been shown to affect abundance of immature turtles [16], previous work compared populations across a much larger geographic scale than our study. Frequent feeding of low-quality food (i.e., bread) by the public to wildlife can often result in disease and poor nutrition [81], which could be responsible for low fecundity, but these potential effects need to be further explored.

Nest predation varied across the study areas with respect to human disturbance. During our surveys, depredated nests were most common in natural and intermediate areas and declined with increasing levels of human disturbance. Although predation rates of simulated nests were low overall, our experimental study lessened concerns about our low detectability of nests in human-disturbed areas and suggests a negative relationship between nest predation and level of human disturbance. This pattern is consistent with studies that suggest that human presence may frighten potential predators [21,82], even though human disturbance can sometimes increase populations of nest predators [83]. The impact of human disturbance on nest predators can vary considerably and may have variable consequences on rates of nest predation. If female reproduction and nesting activity is low in areas of high human disturbance (e.g., city parks), then mammalian predators may not have had the same opportunity to develop a visual or olfactory search image for turtle nests at these locations. These predators can quickly learn how to locate food items when they become abundant [84,85], but if nest abundance remains low at human-disturbed areas, then it is unlikely predators would have been searching for nests during the 72 h period when simulated turtle nests were present during our experiment.

Increased human activity and alteration of landscapes has the potential to substantially change the nesting behavior and ecology of oviparous species [50]. Turtles are particularly vulnerable to changes in their nesting habitat since biotic and abiotic environments within and around nest sites largely determine egg hatching success [86–89], which in turn could impact population demographics. Understanding how human disturbance impacts nesting sites and changes in maternal nesting behaviors will provide important information for predicting how populations of oviparous animals will persist in human-disrupted environments. We show that maternal nest site choice is relatively consistent across areas with different levels of human disturbance, despite some differences in habitat features across our study areas. The general lack of behavioral shifts in human-disturbed areas suggests that preserving natural microhabitat features will be important for populations in areas with human activity. Moreover, limited evidence of reproduction in areas with high human activity is alarming and warrants more research. Future studies that quantify the effects of human disturbance on offspring development within nests [50,90] will provide further insight into how human activities affect recruitment and long-term persistence of populations in anthropogenically modified environments. Overall, our study highlights

an important and relatively understudied aspect of reptile biology (i.e., nesting behavior) that warrants more attention in research programs aimed at understanding the impacts of anthropogenic environmental change. As urbanization and human populations continue to increase, it is crucial to document how urbanization impacts turtle nesting behavior, as well as nesting success across systems, species, and types of human disturbance.

**Author Contributions:** This study was conceived by M.F.C., D.A.W. and M.E.W., with assistance from J.E.L.-P.; M.F.C. and J.E.L.-P. conducted the field work; M.F.C. and M.E.W. conducted the data analyses; M.F.C. drafted the manuscript with edits and revisions from D.A.W., M.E.W. and J.E.L.-P. All authors have read and agreed to the published version of the manuscript.

**Funding:** This study was supported by the Auburn University Intramural Grants Program, NSF grant (DBI-1658694), the Alabama Agricultural Experiment Station and the Hatch program of the National Institute of Food and Agriculture, US Department of Agriculture.

**Institutional Review Board Statement:** This study was approved by the Auburn University Institutional Animal Care and Use Committee (protocol number: 2019-35020).

**Data Availability Statement:** All data and R code are freely available on Zenodo at https://doi.org/10.5281/zenodo.7630636.

**Acknowledgments:** We thank Debbie Folkerts, David Mitchell, Amélie Fargevieille, Josh Hall, Iwo Gross and Andrew Caldwell for their assistance in the field, Mike and Carolyn Williams for access to their property and Larry Lawson and Auburn University's E.W. Shell Fisheries Center for permission to access fisheries ponds. Thanks to A. Wilson who made the Research Experience for Undergraduates Program possible for JELP.

**Conflicts of Interest:** The authors declare no conflict of interest.

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
