# Peer review of "Consistent Nest Site Selection by Turtles across Habitats with Varying Levels of Human Disturbance"

_diversity, doi:10.3390/d15020275_

Round 1

Reviewer 1 Report

I liked this paper and few things to say about it; it is well-written and sound.

Please explain why you have one to three artificial nests created per natural nest – why not same number each time?

 I understand difficulties in finding nests across habitats  (n=43; 4 high, 25 intermediate, 14 low) – consider combining high and intermediate? Or please explain why you did not.

There is a full literature on turtle nests and predation but  few refs that may be worth including:

Foley, S. M., Price, S. J., & Dorcas, M. E. (2012). Nest-site selection and nest depredation of semi-aquatic turtles on golf courses. Urban Ecosystems, 15, 489-497.

Marchand, M. N., & Litvaitis, J. A. (2004). Effects of landscape composition, habitat features, and nest distribution on predation rates of simulated turtle nests. Biological Conservation, 117(3), 243-251.

Dawson, S. J., Adams, P. J., Huston, R. M., & Fleming, P. A. (2014). Environmental factors influence nest excavation by foxes. Journal of Zoology, 294(2), 104-113.

Reviewer 2 Report

The paper “Consistent nest site selection by turtles across habitats with varying levels of human disturbance” is a solid, well thought-out and well executed study on reproductive behavior of emydid turtles. These results can have major implications on the conservation of freshwater turtles, especially in the anthropogenically altered environments. The fieldwork methodology and data gathering are satisfactory, and the authors employ adequate statistical methods. I have only a few remarks which I’ve provided in the annotated .pdf. Notably, some methodological approaches need more clarification – the authors should provide more detail regarding photographing of the canopy, and if they kept the parameters uniform in order to avoid systemic error. Also, in the predation experiment the authors got very low number of cases of predation of the artificial nests. The authors should acknowledge that fact and probably tone down some statements regarding nest predation. Other than that, I think that this paper is a good contribution to the field and I would recommend it to be published in Diversity after a minor revision.

Reviewer 3 Report

The MS deals with a topic of great interest such as the effect of human disturbance on coastal areas, particularly in turtle nesting sites. The document is solidly written and has the necessary foundations in the introductory part. The references are pertinent. I would be glad to see it published in future issues of diversity. Some observations to attend, is:

Remove brackets and semicolons (separate references by hyphen) on line 48
